# A Novel H-2^d^ Epitope for Influenza A Polymerase Acidic Protein

**DOI:** 10.3390/v14030601

**Published:** 2022-03-14

**Authors:** Ida Uddbäck, Jacob E. Kohlmeier, Allan R. Thomsen, Jan P. Christensen

**Affiliations:** 1Department of Immunology and Microbiology, University of Copenhagen, DK-2200 Copenhagen, Denmark; iuddback@sund.ku.dk (I.U.); athomsen@sund.ku.dk (A.R.T.); 2Department of Microbiology and Immunology, Emory University, Atlanta, GA 30322, USA; jkohlmeier@emory.edu

**Keywords:** T cell, T-cell immunity, mucosal immunity, adenovirus, adenoviral vaccines, murine models, influenza vaccine

## Abstract

Understanding the complexity of the T-cell epitope hierarchy in humans through mouse models can be difficult. In particular, using only one murine strain, the C57BL/6 mouse, to investigate the immune response to influenza virus infection limits our understanding. In the present study, by immunizing C57BL/6 mice with an adenoviral vector encoding the polymerase acidic (AdIiPA) protein of influenza A virus, we were able to induce a high number of PA-specific T cells. However, upon challenge, these cells were only partly protective. When instead immunizing BALB/c mice with AdIiPA, we found that the immunized mice were fully protected against challenge. We found that this protection was dependent on CD8 T cells, and we identified a novel H-2D^d^-restricted epitope, PA33. These findings provide a new tool for researchers to study PA-specific immunity in mice with an H-2^d^ haplotype. Additionally, our findings underscore the importance of critically evaluating important limitations of using a single inbred mouse strain in vaccine studies.

## 1. Introduction

Despite available vaccines, the influenza A virus remains a major cause of human disease in today’s society, with up to 3 million cases of severe disease annually worldwide [1]. The variable efficiency of current vaccines is largely due to the high mutation rate of influenza viruses. Point mutations in the genes encoding the surface proteins, hemagglutinin, and neuraminidase may cause conformational changes that prevent antibodies from neutralizing the virus. These changes, termed antigenic drift, give rise to new seasonal strains that force annual evaluation and reformulation of the vaccines, often resulting in vaccines with middle to low efficiency [2,3]. As a result, alternative vaccine strategies against influenza virus are being researched. One way to elicit broad-spectrum immunity is by the generation of memory CD8+ T cells. In difference to a neutralizing humoral response, CD8+ T cells recognize epitopes from conserved internal proteins of the virus. There is evidence in both mice and humans that these cells can help the host recover faster and limit tissue damage after an influenza infection [4,5,6]. We have previously demonstrated that vaccination locally and systemically with an adenovirus encoding nucleoprotein (AdNP) from influenza A virus elicits a long-lasting CD8+ T-cell population that can protect mice against lethal challenge up to at least 8 months post-vaccination [7,8]. However, despite the internal proteins being highly conserved between influenza strains, variations in these proteins still exist. During our studies, we found that challenging commonly used C57BL/6 mice with influenza strains with mutations in the H-2^b^-restricted NP366 epitope resulted in suboptimal protection. Therefore, to be able to generate protection against a broader range of influenza strains, we investigated the potential of including adenoviruses encoding other highly conserved internal influenza genes [9]. One target that has not been extensively researched on its own as a vaccine target is polymerase acidic protein (PA). When aligning influenza PA from 62 sequences, 82% conservation was found, a considerably higher percentage compared to external proteins HA (31%) and NA (34%) [10].

In the present study, we explored the potential of an adenovirus vaccine encoding PA to generate immunity protecting against influenza A virus. Our lab has extensive experience working with adenoviral vectors and has previously found that the addition of MHC-associated invariant chain (Ii) linked to the antigen in the adenoviral vector may enhance the CD8 T-cell response [9,11]. This is especially true where the targeted epitope is not the most immunodominant one. Thus, Ii is included in the adenovirus vector encoding PA (AdIiPA) used in this project. We found that vaccination with AdIiPA induced high numbers of PA224–233 (PA224)-specific CD8 T cells in C57BL/6 mice, but when challenged, only half of the mice were protected from lethal influenza infection. In contrast, AdIiPA-vaccinated BALB/c mice were fully protected against a lethal dose influenza virus, and this protection was confirmed to be dependent on CD8+ T cells. We identified a previously never published MHC I H-2^d^ epitope for influenza PA, PA33–41 (PA33). We confirmed our finding both by intracellular staining for IFN-γ after ex vivo peptide stimulation and through tetramer staining. These findings provide a new tool for exploring PA specific immunity in H-2^d^ mice, but also highlight the limitations of working within only one inbred mouse strain.

## 2. Materials and Methods

### 2.1. Mice

Female mice (C57BL/6 and BALB/c) were purchased from Taconic Farms (Ry, Denmark) and housed in a specific pathogen-free facility. B10.D2 mice were obtained from Jackson Laboratory, Bar Harbor, ME, USA. Upon arrival, all mice were rested for ≥1 week at the facility before being used in experiments. All experiments were conducted in accordance with national Danish guidelines regarding animal experiments as approved by the Danish Animal Experiments Inspectorate, Ministry of Justice (protocol code P20–498 protocol, start date 1 January 2020). The mice were housed in an AAALAC-accredited facility in accordance with good animal practice as defined by FELASA.

### 2.2. Viruses

A replication-deficient adenovirus serotype 5 vector with a deleted E1 region and a nonfunctional E3 gene was used. Inserted in this vector was PA from influenza strain A/Puerto Rico/8/34 linked to the murine invariant chain (designated AdIiPA). AdIiPA was produced according to the previously described vector-generating protocol [12]. AdIiPA particles were purified using standard methods, aliquoted, and stored at −80 °C in 10% glycerol. IiPA insert was confirmed by sequencing (data not shown). AdIiPA stocks were titrated using an Adeno-X Rapid Titer Kit (Clontech Laboratories, Mountain View, CA, USA). All vaccinations were given subcutaneously (s.c.) in the right footpad and intranasally (i.n.) after anesthesia with i.p. injection of avertin (2,2,2 tribromoethanol in 2-methyl-2-butanol, 250 mg/kg). Mice were given a dose of 2 × 10^7^ PFU in 30 uL. Influenza A/Puerto Rico/8/34 was used for challenge studies. For each virus preparation, the lethal dose was determined, and 3 LD50 was used for challenge. Mice to be challenged were first anaesthetized by i.p. injection with avertin, followed by i.n. infection with appropriately diluted influenza virus in 30 uL. Mice were weighed daily after influenza challenge and euthanized by cervical dislocation if the weight loss exceeded 25% or 21 days post-challenge, whichever came first.

### 2.3. In Vivo Depletion of CD8 T Cells

A total of 100 μg α-CD8 mAbs (YTS 156.7.7 and YTS 169.4.2.1) were administered i.p. at days −1, +1, and +4 relative to influenza challenge. Additionally, 5 μg α-CD8 mAbs (YTS 156.7.7 and YTS 169.4.2.1) were administered i.n. at days −1, +1, and +4 relative to influenza challenge after isoflurane anesthesia. Control groups were given isotype IgG (BioXcell) at the same concentrations and at the same time interval. All hybridomas were kind gifts from S. Cobbold, University of Oxford, UK [13]. Cell depletion was verified by flow cytometric analysis.

### 2.4. Influenza MDCK Plaque Assay

Lungs were homogenized using 1% BSA in PBS and sterilized sand with mortar and pestle. Lung suspensions were kept at −80 °C until used. A total of 50,000 MDCK cells per well were seeded in a 96-well plate and grown in complete media (1X DMEM, L-glutamin, SodiumPyruvate, PenStrep, 10% FBS). Then, 24 h later, a 10-fold dilution of lung suspensions influenza growth medium (DMEM 1965 medium with 2 mM L-glutamin, 200 IU/mL penicillin, 50 μg/mL streptomycin, 0.2% BSA, 1% sodium-pyruvate and 5 units/mL TPCK Trypsin) was added to the MDCK cells in duplicate, and they were incubated for 2 h at 37 °C, 5% CO_2_. Next, the virus was removed, and a 1:1 mixture of 2X minimum essential medium (MEM; Eagle supplemented) with 0.4% BSA, 10% NaHCO_3_, 2% streptomycin, 2% penicillin and 5 units/mL TPCK trypsin, and 1.8% methylcellulose was added, and the cells were incubated at 37 °C, 5% CO_2_. Following 48 h incubation, the virus was removed, and cells were fixated with 4% paraformaldehyde in PBS for 30 min at RT. Following fixation, cells were washed once with PBS and permeabilized with warm (37 °C) 0.5% Triton-X in Hanks Balances Salt Solution (HBSS) at RT for 10 min. Next, the monolayer was washed once with PBS, and primary antibody anti-influenza Nucleocapsid Protein (Nordic Biosite) 1:1500 in 10% FBS in PBS was added; then, the mixture was incubated for 1 h, 5% CO_2_, at 37 °C. Following primary antibody incubations, cells were washed with PBS five times, and secondary goat α-mouse HRP-conjugated Ab (Dako) diluted 1:500 in 1% BSA in PBS was added; following this, the assay was incubated for 1 h at 37  °C, 5% CO_2_. Cells were subsequently washed with PBS five times, and substrate solution (3 mg/mL 3-amino-9-ethylcarbazole and 0.07% H_2_O_2_ and 5 mM citrate phosphate buffer, pH 5) was added. Following 30 min incubation at RT, substrate solution was removed, and cells were washed once. Plaque-forming units were calculated using the following formula:

Average number of plaques/well × dilution factor × 20 = PFU/g lung

### 2.5. Preparation of Single-Cell Suspensions and Flow Cytometry

For all analyses of lung cells, intra-vital labelling of circulating lymphocytes was performed. Mice were intravenously injected with fluorochrome-labelled anti-CD3e antibody 5 min prior to anesthesia with avertin followed by exsanguation. Subsequently, bronchoalveolar lavage (BAL) was collected, and lungs, MLN, and spleen were isolated and kept on ice. MLN and spleen samples were passed through a 70 µM filter, washed once, and kept on ice until staining. Lungs were cut into tiny pieces and incubated for 30 min at 37 °C and 5% CO_2_ in digestion media containing 5 g/L collagenase D (Roche) and 1 × 10^6^ U/L deoxyribonuclease I (Sigma-Aldrich, St. Louis, MO, USA) in HBSS. Samples were mixed with a 3 mL syringe every 10 min. Following incubation, cells were washed, and lymphocytes were isolated by the help of a 80/40% percoll gradient and then centrifuged for 25 min at 20 °C. Lymphocytes at the interface were isolated and washed once, then kept on ice until staining. BAL samples were washed once and kept on ice prior to staining. Cells for all experiments were counted on an automated cell counter, Countess II (Invitrogen, Waltham, MA, USA).

For tetramer and surface staining, cells were washed twice with PBS, and Zombie NIR (Biolegend, San Diego, CA, USA) in PBS was added to stain for live/dead cells for 15 min at 4 °C. Cells were subsequently washed with PBS. Next, cells were incubated with tetramer (PA224 tetramer conjugated to PE) in 1% BSA in PBS for 30 min at RT. Antibodies in 1% BSA in PBS staining for indicated surface markers were added, and the mixture was incubated for another 20 min at 4 °C. Next, cells were washed with 1% BSA in PBS twice and were incubated with 1% paraformaldehyde (PFA) for 15 min at 4 °C. After fixation, cells were washed with 1% BSA in PBS and resuspended in 1% BSA in PBS.

For peptide stimulation prior to intracellular cytokine staining (ICCS), spleen, MLN, and BAL samples were incubated in RPMI 1640 cell culture medium (containing 10% FCS supplemented with 2-ME, l-glutamine, and penicillin-streptomycin), 1 μg/mL relevant peptide (PA224–233: SSLENFRAYV or PA33–41: KFAAICTHL), 50 IU/mL IL-2, and 3 μM monensin for 5 h and incubated at 37 °C with 5% CO_2_. Following stimulation with peptide, cells were stained for surface markers. Subsequent to surface staining, the cells were washed and fixated with 1% PFA for 15 min at 4 °C. After fixation, cells were permeabilized with 0.5% saponin in PBS for 10 min at RT. Following permeabilization, cells were stained for intracellular cytokines in 0.5% saponin for 20 min at 4 °C. Next, cells were washed and resuspended in PBS containing 3 μM monensin for analysis.

All samples were analyzed on a LSR Fortessa (BD Biosciences), and data analysis was conducted using FlowJo software V10 (TreeStar, Ashland, OR, USA). The following fluorochrome-conjugated monoclonal rat anti-mouse antibodies were used for surface and intracellular cytokine staining: PE-CF594-conjugated α-CD3e (clone 145–2C11), BV785-conjugated α-CD8a (clone 53–6.7), FITC-conjugated α-CD44 (clone IM7), APC-conjugated α-IFNγ (clone XMG1.2), BV480-conjugated α-CD69 (clone H1.2F3), and BV421-conjugated α-CD103 (clone 2E7). All antibodies were purchased from Biolegend. Relevant tetramers were kindly provided by Søren Buus, Department of Immunology and Microbiology, University of Copenhagen.

### 2.6. Statistical Analysis

Graphpad prism 8 was used for statistical analysis. All samples including 3 or more groups were analyzed using a one-way ANOVA, and pairwise comparisons were performed using the Wilcoxon rank sum test. * denotes *p* < 0.05; ** denotes *p* < 0.01.

## 3. Results

### 3.1. AdIiPA Vaccination Induced Large Numbers of PA224-Specific CD8 T Cells

To investigate the CD8 T-cell response generated by AdIiPA vaccination, C57BL/6 mice were immunized with AdIiPA, and number of IFNγ-producing cells after PA224 stimulation was enumerated in the spleen at 11, 14, 17, 20, 30, and 60 days post-immunization. A peak response was observed at around days 11–17. The number of CD8 T cells was reduced until day 20, and after that, the level of CD8 memory T cells in the spleen remained stable out until day 60 (Figure 1A). The numbers induced, as well as the kinetics, were comparable to what has been observed previously in our studies using adenoviral vectors encoding influenza A nucleoprotein (AdNP) and polymerase basic protein 1 (AdIiPB1) [7,9]. Furthermore, the number of PA-specific memory T cells was analyzed in the airways, lungs, spleen, and mediastinal lymph node (MLN) 60 days after vaccination as well as 5 days after influenza A challenge (Figure 1B). For analysis of CD8 T cells in the lung, an intravascular staining with fluorescent-labelled antibody was administered prior to exsanguination [14,15]. High numbers of PA-specific CD8 T cells were found in all tissues at 60 days after priming. Moreover, the PA-specific T-cell response to a secondary influenza infection was found to be substantially higher compared to unvaccinated groups. However, notably no real expansion of the CD8 T cells in the lungs and airways was observed in comparison to the numbers prior to challenge and 5 days later. Considering the growing number of studies demonstrating the importance of resident memory T cells (TRM) in protection against respiratory infections, the cells in the BAL and lungs were analyzed for canonical residency markers CD69 and CD103 (Figure 1C) [4,16,17]. A high proportion of the CD8 T cells in the airways were expressing CD69 and CD103, demonstrating that AdIiPA vaccination was able to generate a substantial TRM cell population.

### 3.2. Protection by PA-Specific CD8 T Cells Varied between Mice Strains

As described above, immunization of C57/BL/6 mice with AdIiPA generated a substantial population of PA-specific CD8 T cells, both locally and systemically. In our previous studies, we showed that the presence of high numbers of Ag-specific CD8 T cells in the respiratory tract was not in itself sufficient to protect mice against influenza challenge [7,9]. AdNP and AdIiPB1 vaccination can induce similar numbers of antigen-specific CD8 T cells, but these cells, depending on whether they are PB1- or NP-specific, have different in vivo protective qualities. Thus, we next investigated the protective quality of T cells elicited by AdIiPA immunization. In addition, to broaden the scope of this study, challenge studies were performed in two different mouse strains, both C57BL/6 and BALB/c mice. No significant difference in viral titers in the lungs was found in C57BL/6 mice 5 days after challenge between immunized mice and the unvaccinated group (Figure 2A). However, 7 days after challenge, lung viral titers in vaccinated mice were significantly lower compared to unvaccinated mice. Additionally, no difference in weight loss between vaccinated and unvaccinated mice was observed. However, the vaccinated C57BL/6 group had increased survival from a lethal challenge compared to the unvaccinated group (Figure 2B,C). In contrast to what was found in C57BL/6 mice, AdIiPA-vaccinated BALB/c mice had significantly lower viral titers in the lungs already 5 days post-challenge compared to the unvaccinated group (Figure 2D), and by day 7, the vaccinated BALB/c group contained no detectable virus. In addition, AdIiPA-vaccinated BALB/c mice had significantly lower morbidity and mortality after influenza challenge (Figure 2E,F). All vaccinated BALB/c mice survived the challenge and experienced little to no weight loss after a lethal challenge.

The PA-specific CD8 T-cell response in BALB/c mice has not been extensively studied, and to our knowledge, there are no H-2^d^-restricted epitopes for influenza A PA published in the literature. As a result, we were not immediately able to directly study the PA-specific response in BALB/c mice. Moreover, it should be considered that BALB/c and C57BL/6 have different genetic backgrounds and therefore could differ in their immune response. C57BL/6 mice have a skewed Th1 immune response where IFNγ plays a large role, and BALB/c mice have a skewed Th2 immune response with a stronger humoral response. As a first step, we wanted to ensure that the protection against influenza challenge observed following AdIiPA vaccination in BALB/c mice was indeed dependent on CD8 T cells.

To this end, AdIiPA-vaccinated BALB/C mice were depleted of their CD8 T cells and subsequently challenged with influenza virus. Mice depleted of CD8 T cells also suffered a significantly greater weight loss after challenge compared to vaccinated mice receiving isotype control (Figure 3A). Vaccinated BALB/c mice with CD8 T cells depleted both systemically and locally were not able to control influenza virus infection; this was reflected in viral titers in the lungs both at day 5 and day 7 post-infection (Figure 3B). To further investigate if the difference in immunity observed in AdIiPA-vaccinated BALB/c and C57BL/6 mice was dependent on the difference in the general genetic background of the two mouse strains or in the involved MHC restriction elements, we compared the ability to control virus infection between BALB/c mice, C57BL mice, and B10.D2 mice (Figure 3C). B10.D2 mice have a C57BL genetic background with the exception of the MHC alleles, which corresponds to the ones in BALB/c mice. Five days after challenge, both vaccinated B10.D2 and BALB/c mice had significantly lower viral titers in the lungs compared to unvaccinated mice from the corresponding strain. As in our previous experiments, vaccinated C57BL mice showed no increased ability to control the virus infection compared to unvaccinated C57BL mice at 5 days post-challenge.

The above results clearly indicate that CD8 T cells are the major component in the protection of AdIiPA-vaccinated BALB/c mice, but as previously mentioned, no H-2^d^ PA epitope has been published. In order to be able to directly investigate the PA-specific immune response in BALB/c mice, we performed a screen for potentially relevant epitopes using NetMHCpan 4.0 (http://www.cbs.dtu.dk/services/NetMHCpan/ (accessed on 6 December 2019)). After analyzing the T-cell response of vaccinated mice on eight different peptides generated from the screen (data not shown), we found a novel peptide that stimulated IFNγ production in T cells from BALB/c mice 14 days post-immunization (Table 1, Figure 3D). CD8 T cells producing IFNγ after stimulation with PA33 peptide were found in all tissues analyzed: spleen, MLN, and BAL. Using the newly identified epitope, we also confirmed the PA33 specificity using a tetramer (Figure 4A,B). As previously described, we did not observe an expansion of the localized PA population in the challenged AdIiPA-vaccinated C57BL/6 mice, which could in part explain why we did not have optimal protection in these mice. To investigate if there was an expansion of the PA33-specific cells after challenge in BALB/c mice, we enumerated the number of PA33-specific CD8 T cells in BALB/c mice at resting memory phase and after challenge (Figure 4B–D). In contrast to what we observed in C57BL/6 mice, the PA33 population expanded more than 10-fold in both lungs and BAL following intranasal challenge (Figure 4D). Analyzing the expression of residency markers of the PA-specific memory CD8 T cells, a majority of these expressed CD69 and a large proportion also expressed CD103 (Figure 4B), illustrating that the cells induced by the immunization were resident memory T cells.

## 4. Discussion

In this study, we examined the immunogenicity and protective capacities of a new adenoviral vaccine encoding influenza PA antigen, AdIiPA. Priming with AdIiPA induced high numbers of PA-specific CD8 T cells; the PA-specific population was long lived, and a majority of the cells expressed tissue resident markers. There is today ample evidence that T-cell immunity localized to the respiratory system plays an important role in protecting against viral respiratory tract infections; however, one of the key issues to solve is how to maintain a primed T-cell population in the lung environment [18]. The results in the present study confirm and extend previous studies, demonstrating the capacity for adenoviral vectors to persist in vivo and sustain protective immunity locally in the lungs long after administration [8,19,20].

While we found that immunization led to high numbers of airway-associated, antigen-specific CD8 T cells, these cells could only partly protect C57BL/6 mice from a lethal challenge. After challenge, we observed almost no expansion of the PA224 specific population in the lungs and airways. In accordance with previous studies, we hypothesize that despite PA specific cells being generated after priming, the PA224 epitope is not presented locally to a high extent in a secondary challenge with live influenza, and this results in poorer activation and expansion of PA-specific CD8 T cells [21,22,23]. Even though cells specific for multiple epitopes can be induced during priming, the immunodominance can change during a secondary challenge, where PA224 has been shown to be at a disadvantage compared to other epitopes [22]. PA224 is presented mainly by dendritic cells, rather than localized epithelial cells, which delays the response to a secondary infection [21]. This could explain why we saw little or no difference between AdIiPA-vaccinated and -unvaccinated C57BL/6 mice early after challenge (d + 5), while viral titers were significantly lower in the vaccinated mice later in the infection (d + 7). Nevertheless, although a PA vaccine might not be fully effective on its own, it could still be efficient as an add-on vaccine to another antigen where the two T-cell populations could function in synergy.

It should be noted that murine studies investigating influenza vaccines are mainly performed in mice with H-2^b^ haplotype, and alongside highlighting important concepts of epitope hierarchies, it provides us with a limited picture of how to explore and interpret specific vaccine antigen hierarchies in humans [24]. In contrast to what we found in C57BL/6 mice, the PA-specific T cells generated after AdIiPA vaccination in BALB/c mice were fully protective. We identified a novel PA H-2^d^ epitope and showed that not only does the AdIiPA induce PA-specific T cells upon priming in BALB/c mice, but this population is also able to respond and expand rapidly following a secondary challenge. First, this finding offers a new tool for researchers to examine the relevance of PA-specific immunity in H-2^d^ mice. Second, our observations indicate both the potential of a target such as PA and underscore the care that should be exerted when selecting or deselecting targets for vaccine development. While numerous epitopes have been identified in humans, not all of them are highly immunogenic, and while NP and M1 epitopes are the most immunogenic, the top NP and M1 epitopes still only cover about 16–57% of a population [25,26]. This stresses the need to generate a vaccine that can trigger a broad range of T-cell specificities. By combining different relatively conserved targets, one should be able to safeguard against variation in priming efficiency and recall of different antigen-specific populations invariably associated with an outbred population. These variations may occur due to polymorphisms in HLA genes but also reflect variations in the influenza strains, including future upcoming viral mutations.

## Figures and Tables

**Figure 1 viruses-14-00601-f001:**
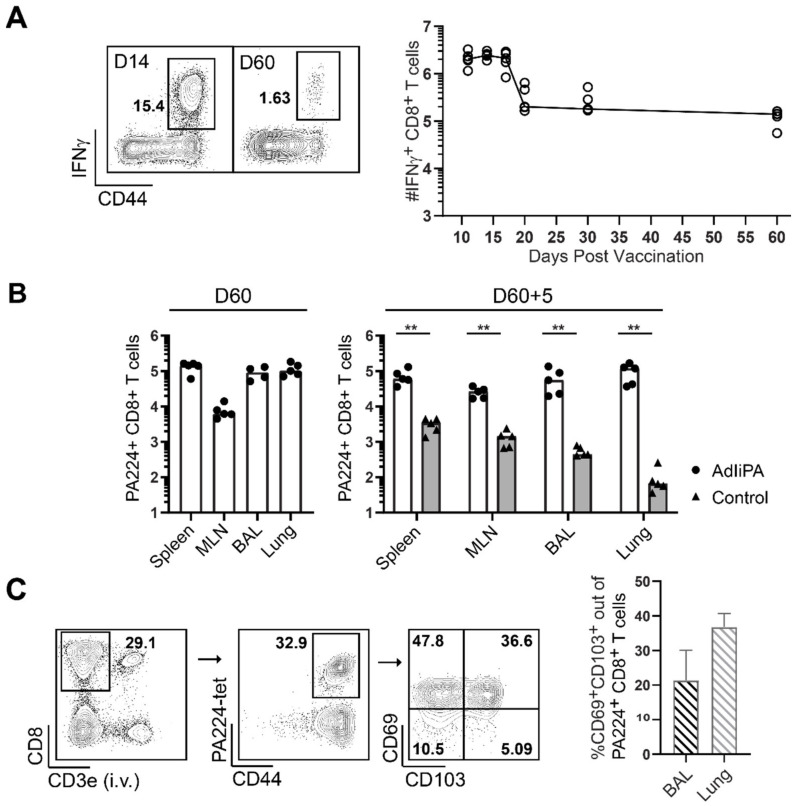
Kinetics of the CD8+ T-cell response to immunization with AdIiPA after priming and challenge. (**A**) C57BL/6 mice were immunized intranasally (i.n.) and subcutaneously (s.c.) with AdIiPA, and spleens were isolated at indicated time points and stimulated with PA224 peptide, followed by ICCS. Samples were analyzed using flow cytometry and enumerated (right). Representative plots of CD8 T cells 14 and 60 days post-immunization (left). (**B**) D^b^-PA224 tetramer-specific CD8 T cells in spleen, mediastinal lymph node (MLN), brochoalveolar lavage (BAL), and lungs were analyzed 60 days after immunization or 5 days after i.n. influenza challenge carried out 60 days after immunization. Controls were naïve mice challenged with influenza. Bars represent mean, and each dot represents one individual mouse. (**C**) Representative plots (left) and percentage of tissue resident T cells (right) out of the PA-specific cells in the lungs and BAL 60 days after immunization with AdIiPA. N = 5 mice. Bars represents mean + SD. Graphs are representatives of experiments repeated three times with similar results. ** = *p* < 0.01.

**Figure 2 viruses-14-00601-f002:**
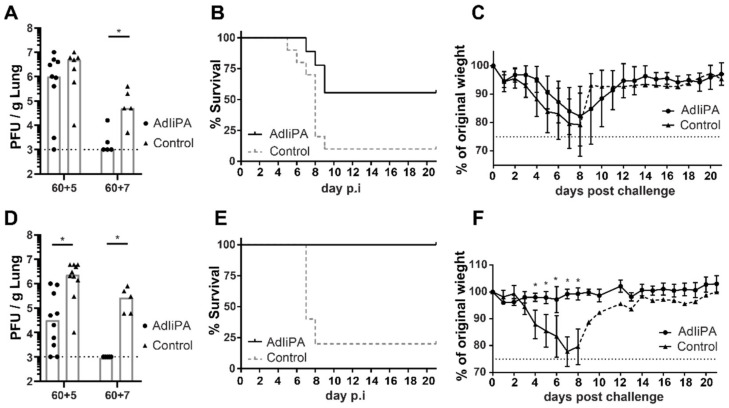
Protection generated by AdIiPA immunization against secondary challenge. C57BL/6 (**A**–**C**) or BALB/c (**D**–**F**) mice were immunized with AdIiPA. 60 days later, mice were challenged i.n. with PR8 influenza. For half of the mice, lungs were isolated at days 5 or 7 post-infection to determine viral titers (**A**,**D**), for remaining mice, survival (**B**,**E**) and weight loss (**C**,**F**) was recorded up until 21 days post-challenge. For weight loss and survival, n in graphs for C57BL/6 mice = 5, and n for BALB /c mice = 5. Graphs are representative of experiments repeated three times with similar results. * = *p* < 0.05.

**Figure 3 viruses-14-00601-f003:**
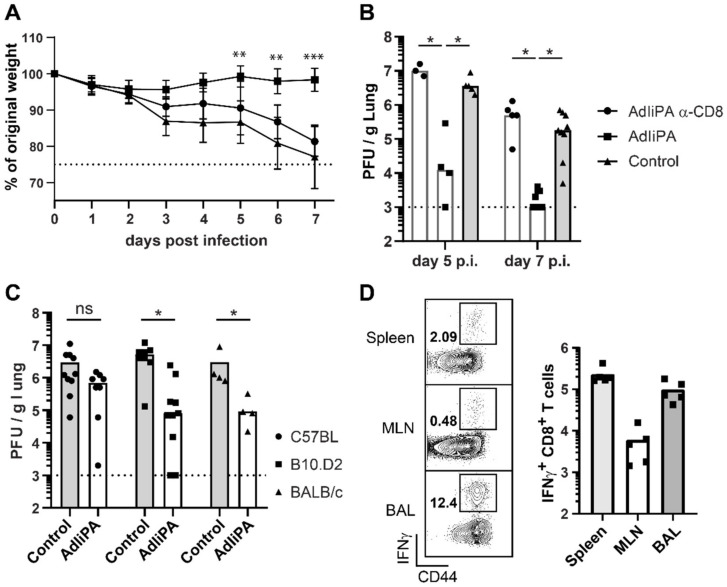
The role of PA-specific CD8 T cells. BALB/C mice were immunized with AdIiPA and challenged i.n. with influenza 60 days later. Indicated groups were given α-CD8 i.p. and i.n. 1 day prior and 1 and 4 days after challenge to deplete CD8 T cells. (**A**) Weight loss of mice was recorded and (**B**) lungs were isolated for titration at days 5 and 7 post-challenge. Asterisks (*) represents statistical differences between AdIiPA and AdIiPA α-CD8 groups. For A: N = 14 mice per group. Data is pooled from 2 replicate experiments (7 mice per replicate). (**C**) Mice with C57BL background but H-2^d^ haplotype, B10.D2, were immunized with AdIiPA and challenged 60 days post-immunization. Viral titers were determined in lungs 5 days post-challenge and compared to BALB/c and C57BL/6 mice. (**D**) Fourteen days after AdIiPA immunization in BALB/c mice, cells from spleen, MLN, and BAL were stimulated with PA33 peptide and ICCS was performed and IFNγ-producing cells were enumerated. Graphs are representative of one experiment repeated twice with similar results. * = *p* < 0.05, ** = *p* < 0.01, *** = *p* < 0.001, ns = non significant.

**Figure 4 viruses-14-00601-f004:**
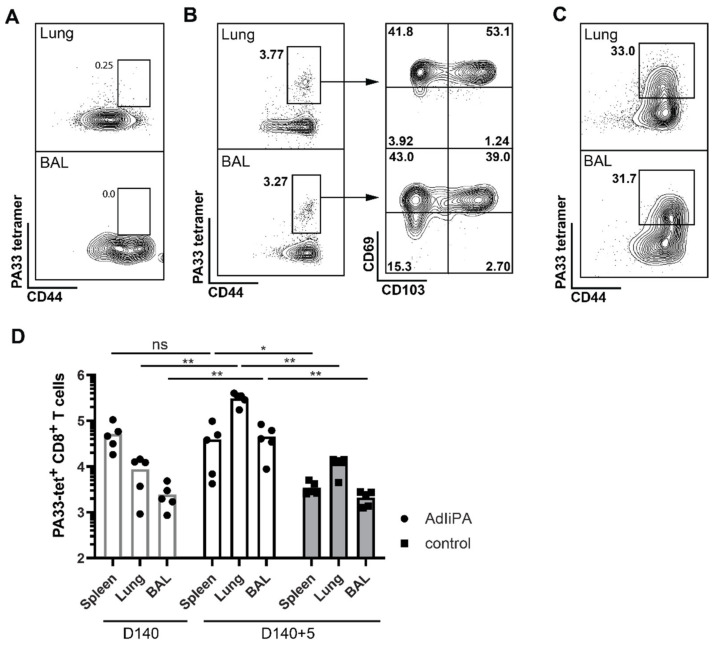
Identification of a novel K^d^ epitope PA33. BALB/C mice were immunized with AdIiPA, and 140 days later, spleens, BAL, and lungs were isolated, and single cells were analyzed by flow cytometry. (**A**) Representative staining of K^d^ PA33 tetramer+ T cells in naïve mice. (**B**) Representative staining of K^d^ PA33 tetramer+ T cells and TRM markers CD69 and CD103 in lung and BAL. These are gated on intravenously stained negative and CD8+ cells. (**C**) Representative plots of tetramer staining in lung and BAL in AdIiPA-immunized mice five days after influenza challenge. (**D**) At 140 days post-immunization, mice were challenged with influenza, and five days post-challenge, spleen, BAL, and lung were isolated and PA33 tetramer+ cells were analyzed by flow cytometry. Total number of PA-specific CD8+ T cells were enumerated by D140 and D140 + 5 days post-challenge. Graphs are representative of experiments repeated twice with similar results. * = *p* < 0.05, ** = *p* < 0.01, ns = non significant.

**Table 1 viruses-14-00601-t001:** Novel H-2^d^ PA epitope.

Position	Restriction	Sequence
PA33–41	K^d^	KFAAICTHL

## Data Availability

Not applicable.

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
