# Peer review of "A Novel H-2d Epitope for Influenza A Polymerase Acidic Protein"

_viruses, 2022, doi:10.3390/v14030601_

Round 1
Reviewer 1 Report
The authors have identified a novel PA epitope important in T-cell immunity from influenza A PR8 in BALB/c mice. While the flow of the paper explained some of the reasonings behind the experiments, there was insufficient detail describing many experiments and a lack of discussion to connect these experiments to human influenza infection.
The biggest concerns include a lack of description of the difference between C57BL and BALB/c mice, which in particular differ in the helper T-cell response. The explanation of using the B10.D2 was well explained but the manuscript would benefit from also stating the differences between C57BL and BALB/c mice. Further, the authors should have examined human MHC binding using the NetMHCpan 4.0 program as they did with the BALB/c mice to identify the 8 peptides tested. Since the use of mice is not the best model for influenza infection and the PR8 strain is a mouse adapted strain, these data are harder to connect to human relevance. If the authors focused on PA peptides overlapping with human MHC binding in the database analysis they performed, this would support relevance to human vaccines and strengthen the manuscript. Further, an analysis of the conservation of the identified PA peptide among different influenza strains, subtypes, and isolates would further strengthen relevance as the authors themselves indicate in the discussion that conserved targets are better to avoid losing immunity as the virus changes.
Other concerns include the lack of details and accuracy in many of the figures. For example, in Figure 1 there is no legend to describe “A” and “B+C” are together, confusing which data is which; further there is no indication of how many mice were used to obtain the data in the Figure 1. It is also unclear if the three trials in Figure 2 had a total of 15 mice (5 in each trial) or a total of 45 mice (15 in each trial). Also, some figures indicate the experiments were done just twice while three replicate trials are standard for peer review publication. Cell percentages throughout appear to be very low, for example ~5% CD8+ T-cells after AdIiPA vaccination. How does this compare to the T-cell response of other vaccines? Data could be strengthened by addition of further controls. For example, in Figure 1 the authors state that there is no increase in T-cell response from prior to challenge to 5 days post challenge; however, there is no control of vaccinated and not challenged. Is it possible a decline in T-cells would have been observed in the no challenge control? Perhaps not, but that data is not presented or possibility discussed. Figure 3 indicates only BALB/c mice in the title, but the figure actually shows data from the different mice strains and should be edited to reflect this. The lack of clear descriptions of the experiments and results made reading the manuscript more difficult and I found myself having to look at external sources for clarity and understanding. To make the manuscript more suitable for a general virology audience more explanations need to be included to support the experiments and interpretation.
Other edits, of less concern, include the use of “acid” versus “acidic” at times for the PA protein and “nucleocapsid” instead of “nucleoprotein”. The correct terms for these influenza proteins are polymerase acidic and nucleoprotein.
Reviewer 2 Report
The manuscript by Uddback et al., tests the protective capacity of an adenovirus vaccine vector encoding the influenza polymerase acid (PA) across two genetically different mouse strains. They show that while immunisation of B6 mice with their AdIiPA vaccine generated sizable pools of PA-specific CD8+ T cells in the lung and circulation these mice were not protected during a high dose influenza virus challenge and this coincided with minimal recall of these PA specific cells. In contrast, immunisation of Balb/c mice resulted in effective protection against viral rechallenge. The authors show that this is CD8+ T cell mediated and go on to identify a novel H2d restricted PA epitope. Overall, this is a well written, straight forward and logical paper that provides interesting insight into epitope selection in influenza T cell-based vaccines.
- Figure 1 – For clarity can the authors specify in the text/figure legends the route they administered the AdLiPA vaccine.
- Figure 1a – It would be helpful to include the number of IFNg+ cells detected in the lung of unvaccinated mice naïve animals so as to highlight the numbers present at d60 are still above background levels.
- Figure 1b + c - It would be useful to merge the data in Fig 1b+c to highlight that the PA specific cells do not expand in the vaccinated mice but they are elevated compared to control animals. This display would make it easier for the reader to understand the statement in line 282 “As previously described, we did not observe an expansion of the localized PA population in the challenged AdIiPA vaccinated C57BL/6 mice, which could in part explain why we do not have optimal protection in these mice”
- Figure 1c – Please provide statistical analysis to support the statement that the number of PA specific CD8+ T cells in the vaccinated group is substantially higher compared to the unvaccinated cohort
- Figure 1 – It is important to add the number of mice per group for each of the bar graphs (or display individual mice)
- Figure 2 - In the control cohorts, are the animals being immunised with an empty adenovirus? If so, how do the viral titres compare to a naïve mouse following challenge (is there any protection mediated by the vaccine vector?)
- Figure 3a – Please provide statistics to support the statement that mice depleted of CD8+ T cells suffered significantly greater weight loss (as stated in line 252).
- Figure 3d – If possible, it would also be useful to show numbers of IFNg+ cells in the lung of Balb/c mice. Once again, the number of mice per group should be listed in the figure legend (or show individual mice).
- Figure 3d – How confident are the authors that the epitope that have identified captures all the PA specific cells in the Balb/c immunised mice? What % of IFNg producing cells are identified when influenza virus infected cells are used as a source of stimulations? Is this comparable to what is detected using the PA33 epiptope?
- Figure 4a – Are the FACs plots in Fig 4a gated on CD8s? If so, it seems as though a large proportion of the memory CD8+ T cells in the BALf of the vaccinated mice are not PA tetramer+ . Is this to be expected? Are these CD8+ T cells reactive to the Ad vector or are the authors missing a proportion of the PA specific response? It would be helpful to see controls showing the level of PA33 tetramer+ CD44+ cells in the BALf of naïve Balb/c mice.
- Figure 4d – Please specify the number of mice per group (showing individual mice would be helpful) and perform the statistical analysis to support the claim that there is a significant expansion in the number of PA33+ CD8+ T cells post challenge.
Minor issues:
- Fig 3a and 3b and 4a are not introduced in chronological order
Round 2
Reviewer 1 Report
The revised manuscript and explanations addressed my concerns. Figure legends are improved and the added text describing the mouse strains is appropriate. It makes sense to limit numbers of mice, and the cell percentages are in line with those expected; although these explanations could be added briefly in the text for a broader scientific audience but perhaps are not necessary for most readers in the field.
Reviewer 2 Report
The authors have addressed all my concerns.